# Epigenetic Changes Associated with Osteosarcoma: A Comprehensive Review

**DOI:** 10.3390/cells12121595

**Published:** 2023-06-09

**Authors:** Luke Twenhafel, DiAnna Moreno, Trista Punt, Madeline Kinney, Rebecca Ryznar

**Affiliations:** 1College of Osteopathic Medicine, Rocky Vista University, Englewood, CO 80112, USA; dianna.moreno@co.rvu.edu (D.M.); trista.punt@co.rvu.edu (T.P.); madeline.kinney@co.rvu.edu (M.K.); 2Department of Biomedical Sciences, Rocky Vista University, Englewood, CO 80112, USA; rryznar@rvu.edu

**Keywords:** osteosarcoma, epigenetics, DNA methylation, histone modification, non-coding RNA

## Abstract

Osteosarcoma is the most common malignant primary bone tumor in children and adolescents. While clinical outcomes have improved, the 5-year survival rate is only around 60% if discovered early and can require debilitating treatments, such as amputations. A better understanding of the disease could lead to better clinical outcomes for patients with osteosarcoma. One promising avenue of osteosarcoma research is in the field of epigenetics. This research investigates changes in genetic expression that occur above the genome rather than in the genetic code itself. The epigenetics of osteosarcoma is an active area of research that is still not fully understood. In a narrative review, we examine recent advances in the epigenetics of osteosarcoma by reporting biomarkers of DNA methylation, histone modifications, and non-coding RNA associated with disease progression. We also show how cancer tumor epigenetic profiles are being used to predict and improve patient outcomes. The papers in this review cover a large range of epigenetic target genes and pathways that modulate many aspects of osteosarcoma, including but not limited to metastases and chemotherapy resistance. Ultimately, this review will shed light on the recent advances in the epigenetics of osteosarcoma and illustrate the clinical benefits of this field of research.

## 1. Introduction

Osteosarcoma (OS) is a malignant primary bone tumor originating from mesenchymal cells that produce an immature bone known as an osteoid [1]. It is the most common malignant bone tumor in children and young adults and has a second peak incidence in the elderly [2]. Primary tumors occur at sites of bone growth, such as the metaphysis of long bones, such as those around the knee and shoulder. A minority of cases originate from the pelvis or locations in the axial skeleton, such as the skull and jaw [2]. OS is typically treated with pre- and postoperative chemotherapy and surgical resection of the tumor, including amputation if necessary [3]. Despite aggressive therapy, OS has a relatively poor prognosis, with a 60–70% 5-year survival rate for localized disease. Part of its poor prognosis is due to OS’s propensity for metastasis. Twenty percent of patients present with metastasis, most commonly to the lungs. For these patients, the 5-year survival rate is a frighteningly low 30% [1]. Despite advances in research, OS survival rates have not drastically improved in 30 years. This lack of progression opens the door for different approaches to OS research and treatment.

In modern medicine, cancer has been intrinsically linked to genetics. Researchers have identified links between various types of cancer and specific mutations, inheritance patterns, and carcinogens. The genetic origins of OS are not fully understood, but most cases are sporadic. Additionally, connections to OS occurrence have been made with radiation therapy, inherited cancer syndromes, such as familial retinoblastoma and Li Fraumeni syndrome, and bone disorders, primarily Paget’s disease [4,5]. Despite advances in OS genetics, the cancer’s heterogeneity and propensity for change have limited the clinical significance of these advances, and no single genetic target has led to a significant decrease in mortality [6]. While genetics has not dramatically changed the way we treat OS, it also does not offer a complete picture of the disease. The complex nature and diversity of phenotypes seen in OS cannot be fully explained by classical genetics.

The emergence of epigenetics has offered a partial solution to the missing information in cancer research. Epigenetics refers to heritable changes in gene expression that occur above the level of the genome, rather than direct modifications to the genetic code. Epigenetic changes that modify gene expression have the potential to change cellular mechanisms and behavior. Aberrant epigenetic changes have ubiquitously been associated with all phases of cancer, including initiation, promotion, invasion, metastases, and chemotherapy resistance [7]. These changes can help explain the large array of phenotypes seen in OS, including highly malignant cells with increased metastatic ability or resistance to treatments. Multiple studies have been performed to profile the epigenetic changes in OS. This active area of research is leading to a better understanding of how these changes contribute to all aspects of the disease and what changes may be leading to various malignant phenotypes. Additional research has examined how OS’s varied epigenome can be targeted therapeutically or used as a prognostic marker clinically [8]. Reviews have discussed advancements in OS research that contribute to the understanding of its pathophysiology, epigenetics, and treatment possibilities, but none to our knowledge incorporate a wide range of epigenetic mechanisms into one complete narrative. The goal of this review is to provide a comprehensive overview of recent advances across multiple modes of epigenetic regulation.

Here, we discuss the epigenetics of OS by breaking it into three broad categories, including DNA methylation, histone modifications, and non-coding RNA (ncRNA). DNA methylation is a process in which a methyl group is added to DNA bases by DNA methyltransferases (DNMTs). In humans, DNA methylation occurs on cytosine-guanine nucleotides (CpG). CpG-dense regions known as CpG islands are often found in the promoter region of genes, where DNA methylation can change the expression level of the gene. In cancer, methylation can either turn on or off the gene [9]. Histones are the primary protein component of chromatin and serve as a scaffold on which DNA is wrapped. Posttranslational modifications of histones change the structure of chromatin, resulting in differential gene expression. Various chemical histone modifications are possible, but the most common are acetylation, methylation, phosphorylation, and ubiquitination [10]. ncRNA includes a wide array of RNA molecules that do not code for proteins but serve various other functions. One example is microRNAs (miRNAs). These small molecules are involved in the post-translational silencing of mRNA molecules. Circular l RNA (circRNA) are single-stranded molecules with various functions, including gene regulation. Long noncoding RNA (lncRNA) is a transcript over 200 nucleotides in length with numerous similarities to protein-coding RNA. lncRNAs have a wide range of functions that relate to gene expression and epigenetic regulation [11]. This review will serve as a comprehensive overview of recent advances across the landscape of epigenetic profiles associated with OS in an effort to improve diagnosis, prognosis, and treatment for OS patients.

## 2. Methylation

Changes in DNA methylation patterns are one of the most common epigenetic modifications seen in OS cancer patients [12]. Transcriptomic studies suggest that the expression profiles of a multitude of genes influenced by methylation state vary in healthy versus OS cells. Some of these specific transcripts are associated with metastatic tumors [13]. DNA methylation is an epigenetic mark that is characterized by the transfer of a methyl group onto cytosine position C5, forming a 5-methylcytosine. As a result, gene expression can be upregulated or downregulated by recruiting gene repression proteins or by inhibiting the binding of a transcription factor [14]. Table 1 provides a summary of the recently described methylation modifications in OS. DNMTs are often responsible for this mark. OS and other cancers that alter DNMT expression can therefore modify their methylation state and epigenetically modify gene expression. Gong et al. illustrate an example of this by showing increased expression of polypyrimidine tract-binding protein 1 (PTB1), positively correlating with DNMTs in nearly all cancers studied, including OS [15]. Studies focusing on how cancers are methylated have been of interest to researchers. In a study written by Parker et al., the methylation levels of five pediatric cancers, including OS, were studied using 309 non-cancerous samples and 489 pediatric tumor samples [16]. A baseline for methylation levels was established for healthy children to identify changes compared to the cancerous samples. Within the 309 healthy samples that were studied, the DNA methylation levels were consistent and maintained that consistency even within diverse tissue types. In the tumor samples, methylation of gene promoter regions was a common finding. They identified 139 genes that were differentially methylated between normal tissue and at least one cancerous tissue. They also found 32 genes that were differentially methylated between the pediatric tumors. The largest difference was found in Spry2, which had a 2.9-fold increase in methylation levels in OS compared to clear cell sarcoma of the kidney. The results highlight that while healthy pediatric tissue maintained a stable methylation level, tumor development significantly altered the DNA methylation state of that tissue [16]. Determining differentially methylated regions in OS patients as compared to healthy patients can provide clinicians with a basis to develop potential biomarkers for diagnosis and assist with personalized medicine treatment plans. This section focuses on changes in methylation that result in altered gene expression and ultimately alter the progression of OS.

DNA is commonly methylated at CpG islands in the promoter regions of genes. In OS, hypermethylation can lead to the silencing of gene expression, allowing for progression or malignant change. Recent work has highlighted the DNA methylation-mediated suppression of miRNAs in OS. Namløs et al., 2022 published a study presenting an example of this form of epigenetic regulation [17]. They previously observed a 3-fold decrease in OS patient samples of miR-486-5p, a miRNA associated with OS cell morphology. Using a microarray, they quantified an increased methylation level on miR-486-5p’s encoding gene, ANK1, in OS cells compared to normal bone samples. When treated with the demethylation agent 5-Aza-2′-deoxycytidine (5-Aza), miR-486-5p expression increased in OS cell lines. Further analysis of OS cells showed hypermethylation of CGI CpG79 upstream of miR-486-5p [17]. In a study by Cheng and Wang et al., 2022, DNA methylation was similarly found to be responsible for the suppression of miRNA [18]. miR-149 was underexpressed in OS samples, and less expression correlated with higher-stage tumors. When miR-149 was transfected into OS cells in vitro, cell growth and metastasis were slowed. Using MSP-qPCR, the study showed hypermethylation at the miR-149 promoter region in cancer tissue when compared to noncancerous adjacent tissue. Further investigation linked the methylation to the specific methyltransferase DNMT3A, which was expressed at higher levels in OS tissue and negatively correlated with miR-149 [18]. A similar pattern was observed by Sun et al. In their study, miR-195 was found to be underexpressed in OS tissue. They also observed corresponding hypermethylation of miR-195’s promoter region. A luciferase assay identified fatty acid synthase (FASN) as a target of miR-195. Underexpression of the miRNA causes increased OS cell proliferation, migration, and invasion capabilities. These effects were reversed with 5-Aza treatment, indicating that hypermethylation of the miRNA promoted malignant traits in the OS cells [19]. A study by Yang et al., 2021 saw a different form of ncRNA, lncRNA, changing the methylation state in OS. qRT-PCR determined that THAP9-AS1 was found to be upregulated in OS cell lines and tissue samples [20]. In the tissue samples, higher levels of THAP9-AS1 expression correlated with a worse clinical prognosis. The results of several assays indicated that THAP9-AS1 could recruit multiple DNMTsnsferases to the promoter region of SOCS3. This theory was strengthened by negatively correlated levels of THAP9-AS1 and SOCS3 mRNA in OS tissues. Additionally, levels of SOCS3 mRNA were recovered in OS tissue when treated with 5-Aza. OS cells also showed higher levels of phosphorylated JAK2/STAT3, which is typically inhibited with SOCS3 expression. These changes in SOCS3 methylation mediated by THAP9-AS1 were correlated with increased OS cell proliferation, migration, invasion, and restrained ROS generation. In mice, THAP9-AS1 overexpression caused increased tumor size and the number of lung metastases [20]. Another lncRNA, HOTAIR, was implicated in OS DNA methylation by Li et al. HOTAIR is highly expressed in OS cells. In HOTAIR knockdown, there was decreased expression of DNMT1 and a decreased level of global methylation. HOTAIR likely facilitates the increased expression of DNMT1 and the following hypermethylated state by inhibiting miR-126. Genes regulated by HOTAIR hypermethylation include CDKN2A [21]. A final lncRNA-related methylation study by Cheng et al. saw the opposite effect: suppression of lncRNA 91 H via siRNA treatment increased methylation of the cell cycle mediator CDK4 promoter. OS cells given this treatment were arrested in the G1 phase. When used in vivo, the treatment decreased tumor size in mice. This was likely due to the previously described methylation, as transfection with CDK4 partially rescued OS cell viability [22]. These studies describe the interplay between ncRNA and DNA methylation in OS. Changes in OS methylation were consistently correlated with more malignant cell traits via the action or inaction of ncRNA molecules.

Changes in OS DNA methylation can directly affect downstream gene products without the intervention of a ncRNA molecule. CXCL12 is a known chemokine that regulates the cell trafficking of leukocytes and tumor cells. Cells expressing the CXCL12 receptors CXCR4 or CXCR7 will migrate by following the CXCL12 gradient. In a study performed by Li et al., CXCL12 expression was evaluated in clinical samples and OS cell lines. Both primary tumors and lung metastases had low levels of CXCL12 and high levels of the receptor CXCR4 compared to normal bone. Methylation-specific PCR confirmed that CXCL12 was hypermethylated in OS cells. DNMT1 knockdown reversed the methylation state, increasing CXCL12 expression in vitro and significantly repressing metastasis in vivo. Suppression of DNMT1 also increased CD8+ T cell responses in lung metastasis, eventually slowing growth and eradicating some tumors. This study showed how a DNA methylation-controlled pathway directly impacted OS metastatic ability and host immune response [23]. In a study by Saeed et al., the human neuronatin gene (NNAT) was also shown to exhibit aberrant methylation of its corresponding CpG island in OS samples. In OS cell lines featuring this methylation state, there was no expression of NNAT, but expression was restored with 5-Aza treatment. Forced expression via transfection resulted in decreased colony formation and transmembrane migration of OS cells. However, NNAT expression did not stop colony proliferation, indicating that suppression of the gene via methylation may be important for the initial establishment of OS tumors. Expression of the gene also played a role in maintaining Ca^2+^ homeostasis and enhancing the cytotoxicity of the SERCA2 inhibitor thapsigargin in human OS cells [24]. These studies provide examples of how the aberrant methylation seen in OS creates measurable differences in downstream gene expression, resulting in changes in the tumor cells and how they interact with their environment.

DNA methylation of tumor-suppressing genes is also facilitated directly by other proteins in OS. SENP3, a sentrin/SUMO2/3-specific protease, is upregulated in OS and associated with reduced expression of E-Cadherin (E-Cad) mRNA [25]. Yang et al. demonstrated that the E-cad promoter region featured abnormal methylation of CpG islands in OS patient samples. The knockdown of SENP3 decreased the methylated base level, as detected via bisulfite conversion. When treated with the methylation inhibitor 5-Aza, OS cells increased expression of E-Cad and reversed the SENP3-induced promotion of invasion, migration, and proliferation [25]. In a different study, hypermethylation of the promoter region of the APCDD1 gene was found in OS cells via the action of DNMT3a. This resulted in reduced expression of the Wnt antagonist [26]. When OS cells were treated with 5-Aza, methylation at the APCDD1 promoter site decreased and protein expression was restored. The DNA methylation-mediated suppression of APCDD1 was associated with increased invasion and metastatic ability of OS cells both in vitro and in vivo [26]. While not all the differences in OS methylation are fully understood, these studies provide examples where the methylation is correlated with a specific protein. These proteins could act as targets to monitor, change, or prevent methylation changes in OS research and treatments.

The epigenetic changes that occur in real OS tumors happen in a dynamic 3D environment with a multitude of factors impacting tumor growth and development. The traditional in vitro models used for OS research cannot fully capture this environment and therefore may not accurately reflect the epigenetic modifications seen in patients’ OS tumor samples. A study by Lin et al., 2022 demonstrated this by generating a 3D bioprinted OS model with OS cells and a shrouding extracellular matrix analog [27]. The group used DNA methylomics and KEGG analysis to compare the methylation states of their 3D model OS cells to those of OS cells grown in traditional in vitro plates and spheroid cultures. Valine, leucine, and isoleucine degradation; autophagy; and adherens junction pathways were among the most enriched pathways, indicating a difference in methylation state. Multi-omic analysis found 276 genes with differently methylated positions (DMPs) and differentially expressed genes (DEGs) between the 3D model and the 2D plates. There were 616 genes with both DMPs and DEGs between the 3D and spheroid cultures, confirming that expression differences in the identified pathways were mediated by changes in methylation [27]. This highlights the importance of how OS methylation is studied. Results from in vitro studies need to be validated with other models or in vivo studies to confirm that detected methylation changes accurately reflect those seen in human OS tumors.

In addition to DNA, RNA molecules can also be modified via methylation. N6-methyladenosine (m6a) is one of the most common modifications made to mammalian mRNA [28]. The methyl group is maintained by a dynamic process where it is added by methyltransferase-like proteins (METTL), also termed “writers” and removed by proteins termed “erasures”. Downstream, other proteins termed “readers” recognize the m6a modification and can promote stability and translation of the mRNA molecule. Recent research has shown that m6a methylation has a role in many types of cancer, including tumor proliferation, growth, invasion, and eventual metastasis [29]. In OS, the writer protein METTL3 has been linked to tumor progression. In a paper by Ling et al., it played a role in the overexpression of DRG1. During embryonic development, the gene DRG1 plays an important role in the process of development. This gene is found in almost all tissues and is downregulated after birth. The function of DRG1 can be regulated by signals, such as hypoxia, DNA damage, androgens, and homocysteine. When expressed beyond normal limits, DRG1 is associated with the occurrence and development of cancer [30]. Across OS cell lines and patient samples, DRG1 was expressed at higher levels than in adjacent OS tissue. High levels of expression were also correlated with larger tumor sizes and an advanced clinical stage in patient samples. DRG1 knockdown led to decreased OS cell viability, inhibited migration, and increased apoptosis rates. Flow cytometry also showed increased cell cycle arrest in the G2 stage. An m6A-IP-qPCR assay showed increased levels of DRG1 mRNA in OS cells. Bioinformatics correlated this with METTL3 expression. METTL3’s influence was confirmed by decreased DRG mRNA stability and m6a levels with METTL3 knockdown [30]. More research regarding METTL3 and OS was performed by Zhou et al., who directly measured the levels of METTL3 expression in various OS cell lines. They then silenced METTL3 to measure its direct effects on OS cells. METTL3 knockdown in most OS cell lines featured lower levels of m6a methylation and decreased colony-forming, migration, and invasion abilities. It also promoted apoptosis in the OS cells. PCR demonstrated several predicted genes were downregulated with METTL3 inhibition, especially the ATPase ATAD2. ATAD2 knockdown had similar results to the METTL3-silenced cells [28]. Taken together, these studies illustrate that m6a methylation has a role in OS tumor progression. The writer protein METTL3 is of particular interest in OS. Further research could show more downstream effects of its upregulation and focus on the protein as a therapeutic target.

**Table 1 cells-12-01595-t001:** Summary of described methylation modifications.

Source	Modification	Modifying Agent	Proposed Downstream Mechanism	Resulting Cellular Changes
[15]	Hypermethylation	PTB1	Increase DNMT expression	Increased cell proliferation, migration, and invasion
[16]	SPRY2 methylation		MAPK/ERK pathway activation	Increased cell proliferation
[17]	ANK1 methylation		Decreased miR-486-5p	increased cell morphology
[18]	miR-149 promoter methylation		NOTCH1-mediated Sonic Hedgehog pathway activation	Increased cell growth and metastasis
[19]	miR-195 promoter methylation		Increased FASN expression	Increased cell proliferation, migration, and invasion
[20]	SOCS3 methylation	THAP9-AS1	JAK2/STAT3 activation	Decreased ROS levels, increased cell proliferation, and metastasis
[21]	CDKN2A methylation	HOTAIR	DNMT1 expression via miR-126 suppression	Decreased cell apoptosis
[22]	CDK4 methylation	lncRNA 91	Cell cycle disruption	Decreased apoptosis, increased cell proliferation, migration, and invasion
[23]	CXCL12 methylation	DNMT1	Decreased CD8+T cell response	Tumor immune response and increased cell proliferation, migration, and metastasis
[24]	NNAT methylation		Bone and Ca^2+^ homeostasis	Thapsigargin sensitivity, increased colony-forming potential, and cell migration
[25]	E-Cadherin methylation	SENP3	Accelerated epithelial-mesenchymal transition	Decreased apoptosis, increased cell proliferation, migration, and invasion
[26]	APCDD1 methylation	DNMT3a	Wnt/B-Catenin singling pathway	Increased invasion and metastasis
[28,30]	Increased m6a levels	METTL3	ATAD2 and DRG1 upregulation	Decreased apoptosis, increased cell migration, and invasion

## 3. Histone Modifications

Histone modification is a dynamic process in which histone-modifying enzymes make post-translational covalent modifications to the histone NH2 terminus, known as the tail. Adding or removing chemical factors from the histone tail alters the structure of chromatin. Broadly speaking, this condenses or decondenses the structure. Heterochromatin, the decondensed form, is transcriptionally active, leading to the expression of local genes, while euchromatin, the condensed form, is transcriptionally inactive, leading to the repression of gene expression. In cancer, the normal balance of active and repressive histone modifications changes the expression of oncogenes and tumor suppressor genes, leading to tumorigenesis [31]. Some of these modifications have been identified and used to discriminate between cancerous and healthy tissue or serve as a prognostic marker in identified cancers [7]. Recently described histone modifications in OS are summarized in Table 2 and Figure 1. 

Histone ubiquitination is one of the least explored forms of histone modification, but it has been implicated in transcriptional regulation and the DNA repair response. Therefore, it is unsurprising that changes in histone ubiquitination are commonly found in various cancers [32]. Recent work has been performed to understand the role of histone ubiquitination in OS. Yadav et al. examined N6-methyadesnoise (m6A) proteins as a regulator of histone ubiquitination in OS. These proteins modify RNA by adding, removing, or reading methyl groups on adenosine in the RNA, ultimately determining the fate of RNA. The group found that the m6A eraser ALKBH5 was expressed at high levels in OS tissue when compared to normal bone samples with immunohistochemical analysis on tissue microarrays. When they silenced this protein with siRNA and shRNA, OS cells showed reduced proliferation, short-term viability, clonogenic growth, and migration in vitro. It also reduced tumor size in a mouse tumor xenograft model. Transcriptome-wide Me-RIP-seq data revealed two proteins, histone deubiquitinase ubiquitin specific peptidase 22 (USP22) and ubiquitin ligase RING finger protein 40 (RNF 40), regulated by ALKBH5. Increased levels of these proteins led to inhibition of histone H2A monoubiquitination and promoted the expression of multiple DNA repair techniques in OS cells [33]. This paper demonstrated how changes in histone ubiquitination can promote the survival and growth of OS and suggested that m6A RNA methylation may be a driver behind some changes in OS epigenetics.

Histone methylation is a common modification that is classified by the addition of a methyl group to lysine or arginine amino acids [34]. Jian et al. investigated the role of histone 3 lysine 27 trimethylation (H3K27me3) demethylation in OS. They found the mRNA of the known H3K27 demethylase, KDM6B, was upregulated in OS biopsies compared to normal tissue samples. mRNA expression levels were even further upregulated in metastatic OS samples. When KDM6B was knocked out in OS cell lines, there was no change in cell growth, but there was decreased migration in transwell experiments, suggesting that H3K27me3 demethylation is involved in OS metastasis. This theory was tested in mice by injecting both control and KDM6B knockdown cells unilaterally into the medullary cavity of their tibia. There was no difference in orthotopic tumorigenesis, but the knockdown mice showed significantly less metastasis to the lungs after three weeks. ChIP-Seq characterized the genomic distribution differences between control and knockdown cell lines, revealing 2298 genes with increased H3K27me3 levels in control OS cells. When combined with transcriptome data, the authors narrowed it down to 37 genes that were downregulated in the control cells. Out of those genes, lactate dehydrogenase A (LDHA) was chosen for further analysis. LDHA’s role in tumor cells has classically been defined as an integral part of aerobic glycolysis by converting pyruvate to lactate. After creating LDHA-knockdown OS cells, this group also noted decreased cell migration and that LDHA overexpression was partly able to rescue the inhibitory effect of KDM6B knockdown on cell migration. This established a link between histone demethylation and OS metastasis through the KHM6B-LDHA axis [35]. Another group, He et al., noted an additional role for H3K27me3 in OS. They saw an upregulation of KDM6B and the related histone demethylase KDM6A in OS samples after treatment with the chemotherapy cisplatin. Patients with OS that were more sensitive to the treatment had higher levels of H3K27me3 that were still intact. The group was able to recreate this in vitro using KDM6A and KDM6B knockdown cells. When treated with cisplatin, the knockdown cell’s colony-forming efficiency declined by 60–90% compared to control OS cells. The knockdown cell colonies also had a greater percentage of cells undergoing apoptosis. They then replicated this in vivo by comparing OS tumor size after treatment with cisplatin or cisplatin and GSK-J4, a selective KDM6A and KDM6B inhibitor. Mice treated with combination therapy had smaller tumors and decreased tumor markers when assessed by immunohistochemistry. To uncover the mechanism at work, RNAseq was used to examine changes in gene expression between the groups. The expression of PRKCA was notably different between the groups, with expression being increased in cells with loss of H3K27me3 [36]. PRCKA is known to have antiapoptotic effects mediated by the RAF/ERK/MAPK cascade and phosphorylation of BCL2 [37,38]. The group concluded that loss of H3K27me3 in OS via the action of KDM6A and KDM6B resulted in increased PRCKA expression, leading to anti-apoptotic and anti-chemotherapeutic effects in the cells [36]. These studies reveal demethylation of H3K27 as a key epigenetic modification in the regulation of many genes in OS.

The previously discussed studies showed histone demethylation in OS via the action of demethylase proteins. Decreased OS histone methylation levels can also be facilitated by a loss of histone methyltransferase proteins. Loss of histone 4 lysine 20 trimethylation (H4K20me3) has been noted as a hallmark and poor prognostic factor in human cancers, but not until the recent work of Piao et al., 2020 has it been associated with OS [39]. For the first time, lower levels of H4K20me3 were seen in both OS tissue samples and OS cell lines when compared to normal samples and osteoblast cell lines, respectively. mRNA expression levels also revealed that there were drastically decreased levels of the methyltransferase SUV420H2 in the OS cell lines. To understand what effect loss of SUV402H2 and H4K20me3 may be having in OS, the group then utilized RNA-seq in SUV420H2 knockdown cells to determine changes in possible downstream targets. The results indicated that SUV420H2 is likely involved in various functions, including cell proliferation, gene expression, and metabolic processes. It may also have a role in the MAPK, P53, TGF, and ErbB signaling pathways [39]. As a recently discovered epigenetic change in OS, the loss of H4K20me3 needs to be fully explored to understand its role in the disease and to discover if it or its downstream effects have any clinical use as a prognostic marker or therapeutic target.

Histone methylation also has a role in non-human OS. Sakthikumar et al. described the mutations found in canine OS across multiple breeds using whole-exome sequencing. After TP53, the second most mutated gene was the histone methyltransferase SETD2, which is responsible for the trimethylation of histone 3 lysine 36 (H3K36me3). While it has not been previously implicated in human OS, SETD2 mutations have been observed in other human cancers where they are related to tumor aggression and metastasis. The study also found that 36% of OS samples had mutations in histone-modifying genes, not including SETD2 [40]. While this study was not conducted with human OS samples, previous work has established overlap between dog and pediatric OS clinically, histologically, and genetically [41]. These findings may make SETD2 mutations a candidate for reevaluation in future OS studies.

Another type of histone modification, phosphorylation, has also been studied in OS. Ying et al. recently uncovered novel histone phosphorylation activities of the Sirt1 protein in OS cells. Sirt1 had previously been studied as a deacetylation and methylation enzyme, but this group noticed that increased levels of the protein were associated with increased levels of histone 3 threonine 3 phosphorylation (H3T3ph) via Western blot. The direct interaction between Sirt1 and H3 was demonstrated in a co-immunoprecipitation study using antibodies against both proteins. Additionally, when WT H3-expressing plasmids were transfected into OS cells with or without si-RNA-mediated Sirt1 knockdown, H3T3ph was only seen in cells with Sirt1. The group also showed the effect of H3T3ph on OS cell autophagy, which had been previously associated with SIRT1. Autophagy is a cell survival mechanism that allows for the recycling of cellular components as a source of energy via the generation of an autophagosome. In OS cells grown under a starvation condition with intact SIRT1, H3T3ph levels, autophagosomes, LC3-1 to LC3-II conversion, and the degradation of long-lived proteins all increased. Additionally, the expression of the autophagy genes ATG5, ATG13, and ATG14 all increased in those cells [42]. Autophagy in OS is often a response that confers resistance to chemotherapy treatments. Further exploring the mechanism discussed here may help us understand why some treatments are ineffective and how to counter the effects of autophagy in OS [43].

### Histone and Methylation Modifications in Cancer Stem Cells

Cancer stem cells (CSCs) make up a population of tumor cells that allow for the renewal and differentiation of cancer cells. These cells have roles in tumorigenesis, progression, and eventually chemoresistance [44]. La Noce et al., 2018 recently worked on what role epigenetic changes, such as histone modifications may have in the CSCs of OS [45]. They treated OS cell lines with the histone deacetylase inhibitor (valproic acid (VPA)) and the demethylating agent (5-Aza). In these cells, the group observed increased expression of the stem cell factors OCT4, NANOG, SOX2, and CD133. They also noticed gross increases toward a stem cell-like phenotype, including increased sarcosphere and colony formation efficiency. Finally, OS cells treated with VPA and DAC demonstrated increased migratory ability in a wound healing assay. To uncover what epigenetic changes were associated with the stem cell phenotype in OS cells, histone modifications and DNA methylation were evaluated by flow cytometry. They found DNA hypomethylation and a rise in H3 acetylation, H3K4me2, and H3K4me3. Additionally, a decrease in H3K9me3 and H3K27me3 was noted. Treated cells also had reduced levels of the histone deacetylase HDAC2 and DNA methyltransferase 3a (DNMT3a). When HDAC2 was knocked out in both cell culture and mouse xenografts, OS cells showed increased stemness and growth [45]. This study profiles the epigenetic changes in OS CSCs and reveals two proteins, HDAC2 and DNMT3a, that are involved in the generation of a stem phenotype in OS. It serves as a base for understanding how OS may begin epigenetically and offers two therapeutic targets that may help curb OS growth and proliferation. A different project by Shen et al. examined the epigenetics of OS in stem cells. This group compared bone marrow mesenchymal stem cells (BM-MSCs) to dental pulp mesenchymal stem cells (DP-MSCs). While BM-MSCs are the cells of origin of OS, there are no reports of dental pulp sarcomas [46]. One possible reason for this and their separate differential potentials may be the regulation of PTEN. The BM-MSCs displayed higher levels of DNA methylation and histone 3 lysine 9 demethylation (H3K9me2) in the promoter region of PTEN. Gene set enrichment analysis further revealed that PTEN-related pathways were also downregulated in BM-MSCs. Western blot showed that the increase in methylation was likely due to higher levels of DNMT3b in BM-MSCs. When that protein was knocked out, PTEN expression increased, and implanted BM-MSCs produced dentin pulp-like structures. Western blot also indicated that the lysine methyltransferase G9a was responsible for the increase of H3K9me2 in BM-MSCs. When G9a was knocked out in vitro, PTEN was upregulated, indicating that G9a is required for the DNMT3B-mediated PTEN suppression found in BM-MSCs. The lab linked these findings to tumorigenesis by attempting to transform both cell types into malignant cells. Following a previously described protocol, BM-MSCs from three individuals were transformed into malignant cells, but DP-MSCs were only transformed after PTEN silencing. Additionally, they analyzed RNA-seq data from normal bone tissue, OS samples, and OS cell lines, finding that both the OS samples and cell lines had significantly downregulated PTEN [47]. Overall, this study also helped shed light on what epigenetic changes with downstream effects could be contributing to tumorigenesis in OS.

**Table 2 cells-12-01595-t002:** Summary of described histone modifications.

Source	Modification	Modifying Agent	Proposed Downstream Mechanism	Resulting Cellular Changes
[33]	H2A monoubiquitination	ALKBH5	Increased USP22 and RNF 40 expression	Increased cell viability, cell proliferation, and migration
[35]	Decreased H3K27me3	KDM6B	LDHA overexpression	Increased metastasis
[36]	Decreased H3K27me3	KDM6A/KDM6B	PRCKA overexpression	Cisplatin resistance and decreased apoptosis
[39]	Decreased H4K20 me3	decreased SUV420H2 expression	Multiple signaling pathways	Not studied
[40]	Modified H3K36me3	*SETD2*		Not studied
[42]	Increased H3T3ph	Sirt1	Increased ATG 5/13/14 expression	Increased autophagy
[45]	Decreased HDAC2 and DNAMT3a	Treatment with VPA and 5-Aza	Increased stem cell factors (OCT4, NANOG, SOX2, and CD133)	Increased stem cell phenotype, cell proliferation, and migration
[47]	Decreased H3K9me2 and increased *PTEN* methylation	G9a and DNMT3B	Suppressed *PTEN* signaling	Malignant BM-MSC transformation

## 4. Non-Coding RNA

ncRNA transcripts are responsible for fulfilling a wide range of functions that modulate gene expression, physiology, and development. Processes involved in gene expression that ncRNA is known to play a significant role in include chromatin remodeling, RNA splicing, RNA editing, assembling macromolecular complexes, translational inhibition, and mRNA destruction [48]. ncRNAs do not encode actual proteins; they do, however, play an important part in determining our complex characteristics. Many ncRNA molecules are directly involved in both gene silencing and activation by influencing the methylation and demethylation of genes, which has subsequently led to a growing focus on epigenetic modifications carried out by ncRNA [49]. The epigenetic-modifying role that ncRNAs play in modulating gene expression in tumor cells is quite substantial and has become a topic of interest within cancer research. Exploration of viable biomarkers in cancer cells, such as specific epigenetic modifications, has become a feasible strategy for developing future cancer therapeutics [50]. Further development of data on ncRNA has simultaneously revealed a large number of undeveloped targets and pathways suitable for epigenetic therapy [50]. Recently described ncRNA in OS is summarized in Table 3.

lncRNAs are characterized as transcripts longer than 200 nucleotides with limited coding potential that are involved in chromatin remodeling, regulation of transcription, and modifications performed post-transcriptionally [51]. These long non-coding transcripts are fundamental regulators of transcription, and recent research has been conducted to further observe the role of lncRNA in OS. Xun et al. investigated the expression and molecular function of lncRNAs by analyzing differentially expressed lncRNAs in OS tumors using Genechip microarrays. The specific lncRNA, lncRNA plasmacytoma variant translocation 1 (PVT1), was identified as a markedly upregulated oncogene in OS [52]. High lncRNA PVT1 expression was correlated with a poorer prognosis, the degree of tumor differentiation, distant metastasis, and disease stage in patients with OS. Further analysis was performed by Xun et al. by knocking down the lncRNA PVT1, which resulted in a reduction in OS tumor cell proliferation, migration, and invasion. Subsequently, knockdown of the lncRNA PVT1 also led to suppressed induction of epithelial-mesenchymal transition (EMT), which is involved in tumor metastasis and invasion [52]. A study by Zi et al. identified another lncRNA associated with OS cell proliferation and migration. However, this molecule, named LINC00619, was downregulated in OS tissue. When overexpressed in vitro, LINC00619 minimized OS cell migration and invasion and induced apoptosis. A possible mechanism for this was proposed when bioinformatics and a dual-luciferase reporter gene assay showed HGF to be a target of LINC 00619. HGF is known to activate the tumor-promoting PI3K/Akt pathway in other cancers. As expected, the group saw a negative correlation between LINC00619 and HGF-PI3K/Akt expression [53]. Another study carried out by Yan et al. further evaluated the upregulated expression and molecular function of the lncRNA PVT1 in OS by exploring what role it plays in the PVT1/mi8R-486 axis. The results of this study suggested that PVT1 promoted OS cell metastasis by sponging miR-486 [54]. Within the OS cell lines studied, miR-496 was found to be downregulated and was shown to be inversely correlated with PVT1 in OS. The research group discovered that by upregulating miR-486, PVT1-induced effects on migration and invasion of OS cells were reversed, which suggests that miR-486 targets PVT1 [54]. This paper provided further insight into what roles lncRNAs play in metastasis and discovered the PVT1/miR-486 axis, providing a new molecular target for treatment of OS. Zhang et al. conducted a research study that examined a different relationship between the lncRNA SERTAD1/2/3 and miR-29c in OS cell lines. The results concluded that the lncRNA SERTAD1/2/3 showed downregulated expression in OS cell lines and was correlated with an unfavorable prognosis, survival rate, distant metastasis, and recurrence [55]. Furthermore, miR-29c was found to be upregulated in OS cell lines as well as promoting OS cell proliferation, invasion, and cell growth. When analyzing the SERTAD1/2/3 and miR-29c axis, overexpression of SERTAD1/2/3 was shown to have a sponging effect on miR-29c by competitively binding to it, thus exerting its anti-OS effect [55]. This study’s findings suggest that the lncRNA molecule SERTAD1/2/3 might play a critical role in OS carcinogenesis.

Aerobic glycolysis is a feature of energy metabolism in cancer cells that increases cancer survival, proliferation, growth, and inhibition of apoptosis. A research study executed by Shen et al. focused their investigation on the role the lncRNA KCNQ1QT1/miR-34c-5p axis plays in regard to the glycolytic enzyme, fructose-bisphosphate aldolase A (ALDOA). It was demonstrated that KCNQ1AT1, a type of chromatin-interacting lncRNA, was upregulated in OS cells and promoted tumor growth by contributing to the Warburg effect, which implies a metabolic shift from oxidative phosphorylation to aerobic glycolysis [56]. Interestingly, it was found that by overexpressing miR-34c-5p, an observed downregulation of ALDOA expression was observed. This was further supported by the findings reflecting KCNQ1AT1 acting as a sponge towards miR-34c-5p by competitively binding to it in OS cells, thus explaining its role in promoting OS growth via ALDOA-mediated glycolysis [56]. This study concluded that the KCNQ1AT1/ALDOA axis may provide crucial information regarding the carcinogenesis of OS, thus contributing to important epigenetic biomarkers that are critical for predicting patient outcomes. Wang et al. found a similar axis interaction between the lncRNA molecule, HCG9, and miR-34b-3p. A relationship between HCG9 and miR-34b-3p was determined when HCG9 knockdown in the OS cell caused enhanced miR-34b-3p expression [57]. HCG was identified to negatively regulate miR-34b-3p by sponging its effects to prevent the induction of apoptosis and cell cycle arrest via downregulation of RAD51, in addition to inhibiting OS proliferation. HCG9 expression was found to be upregulated in OS tissue, as well as being correlated with the pathological features of OS, such as proliferation and invasion [57]. These results provided by Wang et al. revealed additional biomarkers of OS and the theoretical basis of the underlying mechanisms.

Understanding the roles and mechanisms in which lncRNAs participate in OS development provides potential therapeutic targets as well as prognostic biomarkers that may enhance OS patient clinical outcomes. A study coordinated by Lou et al. expanded upon the roles of the miR-32-5p/HMGB1 axis, which found carcinogenic effects exerted by the expression of the lncRNA molecule, HNF1A-AS1, that subsequently influenced the upregulation of the protein, HMGB1 [58]. This study discovered that elevated levels of miR-32-5p induced cell apoptosis and impeded proliferation, migration, and invasion in OS cells. While exploring the role of the miR-32-5p/HMGB1 axis, it was found that miR-32-5p targets the gene coding for the protein HMGB1, which is regulated by the lncRNA HNF1A-AS1, by targeting miR-32-5p. To further support this finding, knockdown of HNF1A-AS1 was shown to enhance apoptosis and impede proliferation, migration, and invasion of OS cells. It was concluded that the lncRNA HNF1A-AS1 could be a potential biomarker for the diagnosis and treatment of OS [58]. Yu et al. saw a similar trend in their study, which analyzed the lncRNA CRNDE and miR-335-3p axis, where CRNDE was shown to regulate miR-335-3p by competitively binding and inhibiting its anti-cancer effects. To further confirm the CRNDE/miR-335-3p axis, it was observed that following the co-transfection of OS cells containing CRNDE knockdown with a miR-335-3p inhibitor, the inhibitory effects of CRNDE were attenuated [59]. CRNDE expression was noted to be strongly associated with poor prognosis, proliferation, migration, and invasion of OS [59]. This study hoped to provide a better understanding of potential molecular targets for the prevention, prognosis, and treatment of OS.

Chemoresistance is a challenge that is often encountered when it comes to finding an effective OS chemotherapeutic treatment. The role of competing endogenous RNA (ceRNA) regulatory networks between lncRNAs, circRNAs, miRNAs, and mRNAs has led to further investigation into their role in chemo-resistance. A study performed by Zhu et al. identified a network between differentially expressed lncRNA MEG3, has-miR-200b-3p(miRNA), and AKT2(mRNA) within chemo-resistant OS cell lines that provides new evidence regarding the mechanisms that contribute to OS multidrug chemo-resistance. Zhu et al. found that the lncRNA MEG3 promotes OS doxorubicin (DXR) resistance by upregulating the expression of AKT2, which could be inhibited by overexpressing miR-200b-3p. Additionally, lncRNA MEG3, as a ceRNA, was observed to sponge miR-200b-3p, which resulted in upregulated AKT2 expression and the identification of its role in OS chemoresistance [60]. These findings provide valuable insight that could uncover some novel targets for reversing the chemo-resistant effects of lncRNA MEG3. Another lncRNA found to play a crucial role in OS chemoresistance was identified by Shen et al. when investigating the epigenetic effects on signaling pathways implicated in the development of Adriamycin (ADM) resistance. Shen et al. identified lncARSR as essential for the development of chemotherapeutic resistance to the chemotherapeutic agent, Adriamycin (ADM). Here, the lncARSR was overexpressed in Adriamycin-resistant cell lines, which promoted not only the viability and migration of OS cells through AKT-dependent pathways but decreased the total amount of apoptosis as well [61]. Upregulation of lncARSR enhanced multidrug resistance-associated protein (MRP1), which proved essential for multidrug-resistant OS development. Interestingly, the OS cells that were Adriamycin-resistant also acquired multidrug resistance against both paclitaxel and cisplatin, which are two other types of chemotherapeutics commonly used to treat OS [61]. On the other hand, Shen et al. discovered that by downregulating lncARSR, the sensitivity to Adriamycin was recovered, which delivers promise for future exploration into lncARSR-targeted regimens to improve clinical outcomes.

A distinctly downregulated lncRNA, Prader-Willi Region Non-Protein Coding RNA (PWRN1), was discovered during a study carried out by Shi et al. and was found to be significantly correlated with advanced-stage metastasis, low survival rates in cancer patients, and chemoresistance. One of the major findings in this study was that cancer patients with low PWRN1 expression had much worse OS than those with high PWRN1 [62]. Additionally, when PWRN1 lncRNA was overexpressed in OS cell lines, the amount of cancer cell proliferation was drastically suppressed, and chemoresistance to cisplatin was markedly reduced. Overexpression was also shown to suppress the in vivo growth of OS xenotransplants. Notably, miR-214-5p (a human microRNA) was identified as a possible ceRNA candidate for PWRN1, and the overexpression of miR-214-5p reversed the anti-cancer effects of PWRN1 on OS cell proliferation and chemoresistance [62].

Huang et al. explored the relationship between histone modification and abnormal lncRNAs in OS and found that the lncRNAs MALAT and SNHG20 were shown to have been greatly overexpressed in OS cell lines. The lncRNA molecule MALAT increased H3K27me3 enrichment in the promoter and distal enhancer, while H3K4me1 and H3K9me3 were shown to be significantly enriched within the downstream region [50]. SNHG20, on the other hand, increased H3K3me3 enrichment in the SJSA1 promoter in addition to H3K36me3 within the downstream region. These results suggest that MALAT and SNHG20 may play an important role in regulating active histones and their expression, which helps uncover highly specific epigenetic biomarkers within OS [50]. Huang et al. also discovered that A2M-AS1, CACNA1G-AS1, LBX2-AS1, and NNT-AS1 lncRNA molecules were downregulated in OS cell lines and happened to be associated with a good prognosis in patients with OS. These results imply that the four lncRNA molecules could actually be protective factors in OS. These four lncRNAs were believed to interact with the screened protein-coding genes through various mechanisms that ultimately shape the pathological progression of OS [50]. A different study performed by O’Leary et al., 2017, with a focus on finding therapeutic biomarkers associated with metastasis-free survival, identified a crucial relationship between the transcription of a lncRNA PARTICLE (promoter of MAT2A) and the WW domain containing oxidoreductase (WWOX) tumor suppressor gene [63]. Transiently elevated expression of the lncRNA PARTICLE was found to influence epigenetic silencing modifications within the WWOX tumor suppressor gene, which significantly reduced WWOX expression and averted FRA16D breakage through fork remodeling/scaffolding [63]. By avoiding this FRA16D breakage that is associated with poor OS clinical outcomes, upregulated expression of the lncRNA PARTICLE was found to be significantly associated with metastasis-free survival in OS patients and has led to the investigation of alternate mechanisms for suppressing WWOX expression [63].

While most OS-related RNA research has focused on lncRNA in recent years, other forms of ncRNA have been the subject of studies as well. MiRNAs are single-stranded molecules 20–23 nucleotides in length that influence gene expression by binding complementary mRNA molecules [64]. Once bound, miRNA cleaves or destabilizes mRNA molecules, inducing gene silencing. MiRNA can influence many genetic pathways, and thousands of publications have linked miRNA dysregulation to tumorigenesis [64]. Work by Hou et al. uncovered a protective role for miR-433-3p against OS. They demonstrated that in OS, miR-433-3p is decoyed by the lncRNA SNHG14. This resulted in the upregulation of the miR-433-3p target, F-box only protein 22 (FBXO22). In OS cell lines, SNHG14 and FBXO22 knockdown resulted in decreased cell proliferation, migration, invasion, and induced apoptosis. These effects were attenuated in SNHG14 knockdown cells when a miR-433-3p inhibitor was added [65]. This study demonstrated how the loss of protective miRNA can affect OS progression. A second study by Ji et al. revealed another protective miRNA molecule. Upregulation of Cadherin-6 (CHD6), a mediator of cell-cell adhesion, has been demonstrated to have an effect on the aggressiveness and metastatic nature of other human cancers, including aggressive thyroid cancer [66]. A histopathologic assay revealed increased CHD6 expression in OS tissue when compared to adjacent noncancerous tissue across 133 OS samples. The level of expression was also associated with stage, tumor size, and patient survival. With a clear association between CDH6 levels and OS, Ji et al. searched for miRNAs that targeted CDH6 expression. Of the several miRNA molecules they uncovered, miR-223-3p showed the greatest ability to decrease levels of CDH6 in OS cells, based on Western blot analysis. Additionally, an invasion assay showed that overexpression of miR-223-3p in OS cells impaired cell invasion and migration abilities. This finding was replicated by Ji et al. in vivo, where OS tumors in mice with overexpression of miR-223-3p were smaller and had less lung metastatic burden. Mice with miR-223-3p-overexpressing tumor cells also had a greater survival probability based on Kaplan-Meier survival analysis. Finally, Ji et al. evaluated miR-223-3p in the 133 OS samples with qRT-PCR. MiR-223-3p was significantly downregulated in the OS tissue when compared to the adjacent non-cancerous tissue. As expected, miR-223-3p was also correlated with tumor size, stage, and patient survivability [66]. When another miRNA, miR-451, is underexpressed in cancers, tumors are thought to show more aggressive behavior and upregulation of the AKT/mTOR signaling pathway. Cao et al. showed that miR-451a overexpression in OS cells led to decreased colony number, decreased migration ability, and an increased apoptosis rate in vitro. With Western blots of AKT/mTOR proteins, they showed that miR-451a can inhibit the pathway by preventing the phosphorylation of AKT by PDPK1. However, miR-451a and PDPK1 had low-predicted interactions. Through a series of studies, the group found miR-451a’s target to be YTHDC1, which upregulates PDPK1 via m6A methylation. This mechanism was validated in vivo, where miR-451a transfection inhibited tumor growth and induced cell apoptosis in mice [67]. These miRNA molecules could be the focus of future OS therapies that either target miRNA or increase expression in order to decrease the aggressive and pro-metastatic traits of tumor cells.

circRNA also has a role in OS tumorigenesis and progression. CircRNAs are single-stranded RNA molecules featuring a covalent bond between their 5′ and 3′ ends, giving them a loop-like structure. This unique structure confers exonuclease resistance to circRNA, making them more stable than their linear counterparts [68]. CircRNAs have a wide array of functions, ranging from miRNA-binding sponges to the regulation of gene transcription and translation. With its overall effects on gene expression, it is unsurprising that circRNA has been implicated in numerous types of cancer. Studies have also shown circRNA’s promise as a cancer biomarker [68]. Shen et al. specifically investigated the role of circRNAs in the increased aerobic glycolysis seen in OS cells. Using an RNA immunoprecipitation (RIP) microarray, they identified 10 circRNAs bound to c-Myc, a transcription factor implicated in the expression of glycolytic genes [69]. CircECE1 was the most enriched signal. They found CircECE1 expression increased in OS cell lines and in OS lung metastasis samples. In OS cell lines, CircECE1 knockdown decreased proliferation/migration and increased the apoptosis rate. Overexpression had the opposite effect and increased the colony-forming ability of OS cells. In the CircECE1-overexpressing cells, RT-qPCR and Western blot showed upregulation of several c-Myc targets. This was reversed by a c-Myc knockdown, indicating that CircECE1 regulation of c-Myc contributed to the change in cell characteristics. Shen et al. then linked CircECE1 overexpression to the Warburg effect in OS cells. Overexpressing cells had increased levels of several glycolysis genes, pyruvate dehydrogenase, and ATP production levels. These changes were reversed when C-myc was knocked down in the CircECE1-overexpressing cells. RNA-seq identified the gene TXNIP as a downstream target of increased CircECE1 expression. Expression of TXNIP was decreased by both CircECE1 and c-Myc. Experimental upregulation of the protein reversed the changes in OS growth, proliferation, and glucose metabolism associated with CircECE1. Finally, the group observed similar results in vivo, where CircECE1 overexpression increased OS tumor size, lung metastases, and the expression of glycolytic enzymes [69]. CircECE1 could serve as a future treatment target or biomarker for OS.

The more common malignancy-promoting mechanism of circRNA in recent OS literature suggests it is functioning as a competitive endogenous RNA. Most frequently as a sponge, binding up miRNA molecules and preventing their downstream effects. Yang et al. identified increased expression of circRNA as a mechanism of action. In 40/55 OS samples, circ_001422 expression was increased when compared to noncancerous osteoblasts and correlated to increased tumor size, stage, and metastases. In vitro, knockdown of circ_001422 decreased OS DNA synthesis and colony-forming ability. Flow cytometric analyses showed that knockdown of circ_001422 caused G0/G1 arrest and increased cell apoptosis [70]. When mice were inoculated with the circ_001422 knockdown cells, they had smaller tumors and fewer lung metastases compared to control OS mice. The group used bioinformatic databases and a pull-down assay to uncover that circ_001422 binds and decreases levels of miR-195-5p, the same miRNA suppressed by hypermethylation in the work by Sun et al. [19]. PCR confirmed that levels of the miRNA were reduced in OS samples. qRT-PCR and Western blot analysis were used to find the downstream effects of the circ_001422-miR-195-5p interaction. The overexpression of circ_001422 in OS cells increased expression of the oncogene FGF2 and phosphorylation of PI3K and Akt. These changes were correlated with the increased malignant characteristics of circ_001422 overexpression and reversed by exogenous miR-195-5p [70]. A similar study by Li et al., 2020 found the circRNA, cir-ITCHI, highly expressed in OS cell lines [71]. Overexpression was also associated with increased OS growth, migration, and invasion. Similarly, they showed that overexpression of cir-ITCHI was associated with inhibition of a miRNA. The silencing of cir-ITCHI increased levels of the miRNA (miR-7) and reversed the malignant changes in OS cells. Using Western blot, it was observed that cir-ITCHI-mediated silencing of miR-7 increased activation of the EGFR pathway, ultimately promoting metastasis and growth of OS cells [71]. While several studies showed that overexpression of circRNAs promotes OS growth, they can also be protective against tumor proliferation while acting as a miRNA sponge. Liu et al., 2021 found 20 circRNAs that were downregulated in OS, with the most dramatic being the downregulation of circMTO1 [72]. Transfecting OS cell lines with circMTO1 inhibited migration and invasion of OS cells. Flow cytometry analysis also showed a higher percentage of apoptosis in the transfected cells. The group then determined that out of potential circRNA-miRNA interactions, the miRNA miR-630 was overexpressed in OS and its downstream target KLF6 mRNA was decreased. Experimental overexpression of circMTO1 raised levels of the tumor suppressor gene KLF6. They concluded that in OS, loss of circMTO1 increases the activity of miR-630, leading to the loss of activity of KLF6 and promoting tumor proliferation [72]. CircRNA molecules that act as miRNA sponges could exert a lot of control over gene expression in OS. Future studies that investigate the roles of miRNA in OS should consider circRNA as a potential cause of differing miRNA expression levels.

**Table 3 cells-12-01595-t003:** Summary of described non-coding RNA modifications.

Source	Modification	Proposed Downstream Mechanism	Resulting Cellular Changes
[50]	Increased lncRNA MALAT and SNHG20	Histone modifications	Not studied
[52,54]	Increased lncRNA PVT1	Induction of epithelial-mesenchymal transition and miR-486 suppression	Increased metastasis, cell proliferation, and migration
[53]	Decreased LINC00619	HGF-mediated PI3K/Akt pathway activation	Decreased apoptosis, increased cell migration, and invasion
[55]	Decreased lncRNA SERTAD 1/2/3	Increased miR-29c activity	Increased metastasis
[56]	Increased lncRNA KCNQ1AT1	Decreased miR-34c-5p-meditated ALDOA	Aerobic glycolysis
[57]	Increased lncRNA HCG9	Decreased miR-34-3p activity	Decreased apoptosis and cell cycle arrest, increased cell proliferation, and invasion
[58]	Increased lncRNA HNF1A-AS1	Increased HMGB1 expression via miR-32-5p suppression	Decreased apoptosis, cell proliferation, migration, and invasion
[59]	Increased lncRNA CRNDE	Decreased miR-335-3p	Increased cell proliferation, migration, and invasion
[60]	Increased lncRNA MEG3	Increased AKT2 expression via miR-200b-3p suppression	Doxorubicin resistance
[61]	Increased lncRNA ARSR	MRP1 expression	Multidrug resistance
[62]	Decreased lncRNA PWRN1		Chemoresistance and increased cell proliferation
[63]	Increased lncRNA PARTICLE	Decreased WWOX expression	Not studied
[65]	Decreased miR-433-3p	Increased FBXO22 expression	Decreased apoptosis led to increased cell proliferation, migration, and invasion
[66]	Decreased miRNA-223-3p	Increased CHD6 expression	Increased cell migration and invasion
[67]	Decreased miR-451a	AKT/mTOR pathway activation	Decreased apoptosis, increased migration, and colony-forming ability
[69]	Increased CircECE1	Interacts with C-Myc to inhibit TXNIP transcription	Warburg effect activation, increased cell proliferation, and metastasis
[70]	Increased circ_001422	Increased FGF2 and PI3K/Akt phosphorylation	Decreased apoptosis, increased proliferation, and metastasis
[71]	Increased cir-ITCHI	EGFR pathway activation via miR-7 silencing	Increased metastasis and cell growth
[72]	Decreased circMTO1	KLF6 suppression via increased miR-630	Decreased apoptosis, decreased cell migration, and invasion

## 5. Treatments

An in-depth investigation of the molecular processes governing the development of OS can prove useful in identifying biomarkers for personalized medicine. In particular, the determination of epigenetic modifications correlated to individualized patient OS progression and metastasis can lead to improvements in prognosis and therapeutic efficiency. A highlighted selection of epigenetic modifications and pathways for OS therapy and prognosis are illustrated in Figure 2. 

Specific epigenetic modifications have been revealed as prognostic biomarkers for OS. Using the Therapeutically Applicable Research to Generate Effective Treatments database (TARGET), profiles of EMGs (epigenetic modification-related genes) and lncRNAs were identified. After investigating the EMGs and their clinical significance in OS, a nomogram was then developed for assessing prognosis in OS patients [73]. The four EMGs identified included the well-characterized genes MYC, TERT, EIF4E3, and RBM34 [73]. Consulting the same TARGET database, Yu et al., 2021 found that five lncRNAs (RP11-128N14.5, RP11-231/13.2, RP5-894D12.4, LAMA5-AS1, and RP11-346L1.2) served as a reliable prognostic signature for OS patients with an AUC (area under the curve) prediction for the 5-year survival rate at a 0.745 accuracy [74]. Both of these studies offer promise for improving the prognoses of OS patients.

DNA methylation has proven to be a useful marker in developing prognostic and classifying algorithms for OS. Broadly, Lietz et al. studied the genome-wide methylation state of primary OS tumors. They found that the level of methylation was strongly correlated with patient outcomes. Hypomethylated samples had better outcomes and an improved response to standard chemotherapy treatments. The methylation patterns they observed were reproducible in three other data sets [77]. The use of this algorithm could help clinicians’ decision making on OS treatment options. Another study used multi-omics data to assess how changes in DNA methylation and transcription of immune-related genes changed the OS tumor microenvironment and overall patient prognosis. They categorized patient samples based on immune-related DNA methylation patterns (IMPs) and machine learning and created an associated prognostic risk model. Their model was correlated with differing drug sensitivities and prognoses based on the accompanying tumor microenvironment [75]. Barenboim et al., 2021 used a DNA methylation probe-based algorithm to divide OS patient samples into groups based on their “BRCAness.” BRCAness was defined as a phenotypic trait in tumors with defects in homologous recombination repair resembling inactivation of the BRCA1/2 genes [78]. Here, BRCAness-positive tumors had a greater degree of genomic instability. Gene set enrichment analysis of their BRCAness-positive group detected enrichment of DNA replication, mismatch repair, and homologous recombination signatures. They validated their algorithm with an independent OS sample set with an accuracy of 90% [78]. BRCAness in tumors makes them susceptible to ADP-ribose polymerase inhibitors (PARPi) [79]. DNA methylation analysis may help stratify patients prognostically and contribute to clinical decision-making and treatment options.

Once specific epigenetic modification-related genes associated with OS prognosis and progression are identified, gene-specific epigenetic treatments can be developed. Lu et al., 2020 discovered epigenetic-modifying enzymes responsible for activating a SE (super enhancer) in OS progression [80]. LIF, or leukemia-inhibitory factor, is an essential factor under the control of an OS-specific SE. The histone 3 lysine 27 tri-methylation (H3K27me3) demethylase UTX was identified as an activator of LIF transcription in cases of OS [80]. In OS cell lines, GSK-J4, which is a UTX inhibitor, resulted in an increase in H3K27me3 and a decrease in histone 3 lysine 27 acetylation (H3K27ac) at the LIF gene locus. As a result of this treatment, OS cells displayed a defect in stem cell-like characteristics through changes in histone acetylation associated with genes participating in NOTCH signaling pathways [80]. Another research group suggested epigenetic modification of the gene ZEB1, whose overexpression in OS blocks differentiation and increases metastatic colonization capacity [81]. An imprinted gene locus regulating expression of ZEB1 is the DLK1-KIO3 locus, which encodes miRNAs that target ZEB1. In OS, this locus is hypermethylated, causing a drastic decrease in the expression of these ZEB1 modifiers. When OS cells were treated with the epigenetic drug that inhibits DNA methylation, 5-Aza, this resulted in an increased expression of the miRNA regulators and downregulation of ZEB1, indicating a therapeutic potential for a ZEB-1-driven OS [81].

Other research studies have identified particular epigenetic-modifying enzymes targeting genes involved in OS pathology. ALKBH5 (or ALKB homolog 5) removes N6-methyladenosine from RNAs and has been found to inhibit human OS tumor cell growth, migration, and invasion through epigenetic modification of pre-miRNA-181b-1 and YAP (yes-associated protein) [82]. Based on this result, ALKBH5 can be considered a tumor suppressor gene and therefore could be overexpressed for the treatment of OS [82]. Another gene involved in OS progression is VEGI, or vascular endothelial growth inhibitor. OS cells treated with the epigenetic modifying drugs sodium valproate (VPA) and hydralazine hydrochloride (Hy), with the mechanisms of action of a histone deacetylase inhibitor and a DNA methylation inhibitor, respectively, resulted in the expression of VEGI and DR3. DR3, or death receptor 3, is one of the two receptors by which VEGI is active [83]. The researchers for this study suggested DNMTs and histone deacetylase inhibitors can induce expression of tumor suppressor genes in OS and exert anti-angiogenic effects [83].

Lastly, some drugs explored for their potential to treat OS show mechanisms of action, at least in part, through changes in epigenetic modifications. For example, Xiong et al. showed that treatment of OS cells with GlaxoSmithKline 343 (GSK343) inhibited expression of the histone methyltransferase (enhancer of zeste homolog 2 (EZH2)). Inhibition of EZH2 resulted in apoptosis and autophagic cell death in OS cells. Decreased EZH2 also resulted in decreased expression of c-Myc and the c-Myc regulator FUSE binding protein 1 (FBP1) [84]. Another group was also able to reduce EZH2 expression with the plant alkaloid berberine. In their study, OS cells treated with berberine resulted in reduced cell viability, colony formation, wound healing ability, and migration [85]. By modifying epigenetic regulation in OS cells, these studies show how therapeutics could control c-Myc and decrease proliferation in malignant cells. A final study showed that the nanoparticle-delivered drug combination of gemcitabine and epirubicin resulted in a 250% reduction in tumor volume, and associated epigenetic changes noted were miR-21 and miR-10b decreasing in expression while tumor suppressor-associated miR-34a increasing in expression levels [76]. Table 4 provides a full summary of discussed treatments and models. 

## 6. Conclusions

In this review, we examined the epigenetic changes seen in OS. Different epigenetic modification pathways, including DNA methylation, histone modifications, and non-coding RNA molecules, all exert control over various and sometimes overlapping signaling pathways driving tumor formation or progression in OS. Broadly, these changes make OS more malignant. Across many studies, aberrant epigenetic changes resulted in the enhancement of OS proliferation, migration, invasion, and decreased cell apoptosis. However, in vivo studies where epigenetic-mediated changes were reversed or inhibited resulted in tumor size reduction. These exciting results indicate that epigenetic modifications and markers are promising therapeutic targets for treating OS. Additionally, further study of the epigenetic markers in OS will lead to the continued development and improvement of prognostic models for patient tumors. These models can help inform patients and their physicians at the onset of diagnosis and in the development of care plans. As the understanding of epigenetic modifications and the many pathways they govern in OS continues to advance, there is an opportunity to improve current diagnostic, prognostic, and therapeutic options, ultimately leading to better patient outcomes.

## Figures and Tables

**Figure 1 cells-12-01595-f001:**
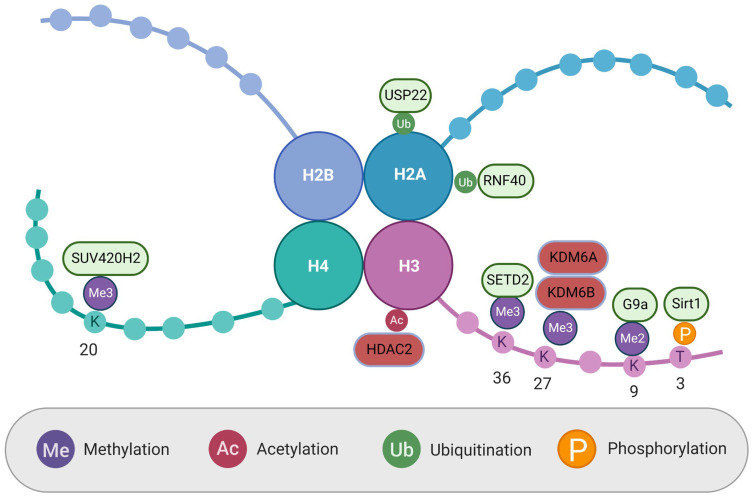
Histone modifications and regulating proteins. Proteins in green add corresponding histone modifications. Proteins in red remove corresponding histone modifications. Created with BioRender.com.

**Figure 2 cells-12-01595-f002:**
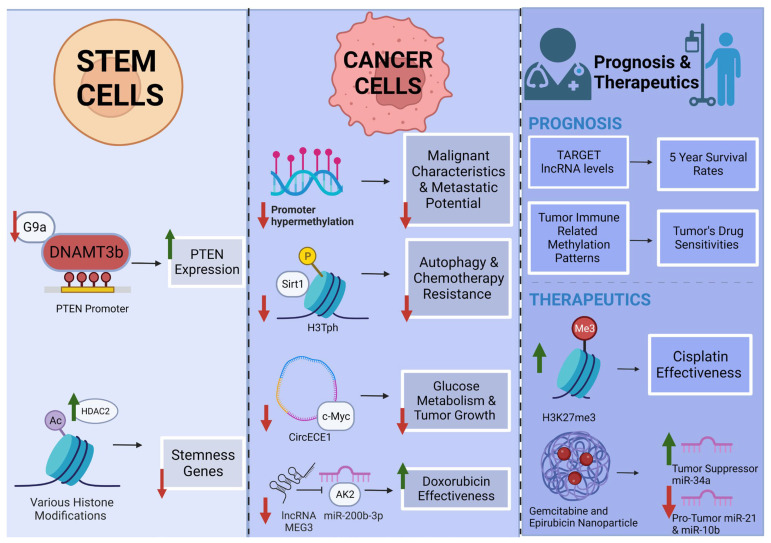
Highlighted epigenetic modifications and pathways for osteosarcoma therapy and prognosis A. Stem cells: Inhibition of G9a decreases methylation of the PTEN promoter, increasing PTEN expression [47]. HDAC2 acts via multiple histone modifications to decrease stemness in CSC [45]. B. Cancer Cell: Decreased methylation in the promoter region of various tumor suppressors decreases malignancy characteristics. Inhibition of Sirt1-mediated H3Tph decreases OS cell autophagy and chemoresistance [42]. Inhibition of CircECE1 prevents c-Myc-mediated growth [69]. Inhibition of MEG3 prevents inhibition of miR-200b-3p, increasing doxorubicin effectiveness [60]. C. Prognosis and therapeutics: target lncRNA and immune-related models [73,74,75]. Cisplatin with GSK-J4 increases H3K27me3, leading to increased drug effectiveness [36]. Nanoparticle delivery of gemcitabine and epirubicin modifies levels of tumor-related miRNAs [76]. Created with BioRender.com.

**Table 4 cells-12-01595-t004:** Summary of described treatments and models.

Source	Treatment	Method of Action	Effect on Tumor
[73,74]	TARGET database	Prognostic Model	EMGs and lncRNA levels used to assess patient prognosis and the 5-year survival rate
[75]	Immune-related DNAmethylation patterns	Prognostic Model	Methylation patterns associated with drug sensitivities and patient outcomes
[76]	Gemcitabine and epirubicin	Nanoparticle drug delivery	Changes in miRNA expression levels leading to decreased tumor volume
[77]	Genome-wide methylation state	Prognostic Model	Tumor hypomethylation associated with better patient outcomes and a greater response to chemotherapeutics
[78]	DNA methylation probe	Prognostic Model	Probes used to categorize tumors based on the “BRACness” phenotype and assess susceptibility to PARPi
[80]	GSK-J4	UTX inhibitor	Increased H3K27me3 and decreased H3K27ac at LIF, resulting in a defect in stem-cell-like characteristics
[81]	5-Aza	Inhibits DNA methylation	Increased expression of miRNA regulators and decreased expression of ZEB-1
[82]	ALKBH5	Modification of pre-miRNA-181b-1 and YAP	Decreased tumor cell growth, migration, and invasion
[83]	VPA and Hy	Histone deacetylase inhibitor and DNA methylation inhibitor	Induced expression of VEGI/DR3
[84]	GSK343	Histone methyltransferase inhibitor	EZH2 inhibition resulting in induced apoptosis and autophagic cell death
[85]	Berberine	c-Myc modification	Decreased cell proliferation

## Data Availability

No new data was created or analyzed in this study. Data sharing is not applicable to this article.

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
