# Peer review of "Epigenetic Changes Associated with Osteosarcoma: A Comprehensive Review"

_cells, 2023, doi:10.3390/cells12121595_

Round 1
Reviewer 1 Report
In this review article Twenhafel et al. summarize the epigenetic alterations of osteosarcoma. This review is informative and useful as a dictionary to wide variety of readers. Especially, the contents regarding non-coding RNA is unique and unprecedented. Therefore, the manuscript seems to be acceptable for publication as it is.
Author Response
Thank you for reading our paper! We appreciate your time to time and are glad you enjoyed it.
Reviewer 2 Report
The review nicely presents many relevant informations regarding various epigenetic modifications that play role in pathophysiology of osteosarcoma. It is very well written, with few minor errors. On line 502 there should be a dot at the end of the sentence, before HCG. The same at the line 525. At the line 780, there is al. missing in Xiong et al. It can be published after thorough proof-reading and these minor corrections.
Author Response
Thank you for the feedback on our paper. We have made the suggested changes.
Point 1: Edits at lines 502,525, and 780
We corrected the writing errors noted at each line.
Point 2: Thorough Proofreading
We have made another pass through the entire paper making minor writing edits and modifying word choice.
Reviewer 3 Report
This is a very interesting and welcome review at a time like this. It provides an overview of epigenetic knowledge in osteosarcoma, which is absolutely essential at the present time.
The authors should revise the wording a little to avoid repetition and lighten the text a little. Apart from these remarks, the review is well structured and deserves to be published.
Author Response
Thank you for reading our paper! We appreciate your feedback on it and have made changes based on your comments.
Point 1: "The authors should revise the wording a little to avoid repetition and lighten the text a little. Apart from these remarks, the review is well structured and deserves to be published."
We have gone over the paper again making edits focused on your comment. We have reduced the number of introductory clauses and unnecessary words throughout the text in order to help lighten it. In addition, we have made an effort to vary commonly used language such as descriptors like "increased".